# Green Propolis Compounds (Baccharin and p-Coumaric Acid) Show Beneficial Effects in Mice for Melanoma Induced by B16f10

**DOI:** 10.3390/medicines8050020

**Published:** 2021-04-30

**Authors:** Gabriel H. Gastaldello, Ana Caroline V. Cazeloto, Juliana C. Ferreira, Débora Munhoz Rodrigues, Jairo Kennup Bastos, Vanessa L. Campo, Karina F. Zoccal, Cristiane Tefé-Silva

**Affiliations:** 1Centro Universitário Barão de Mauá (CBM), Rua Ramos de Azevedo, n 423, Ribeirão Preto, SP 14090-180, Brazil; gabrielhenriquegastaldello@gmail.com (G.H.G.); caroline.cazeloto@hotmail.com (A.C.V.C.); julianacogo_@hotmail.com (J.C.F.); vanessa.campo@baraodemaua.br (V.L.C.); karina.zoccal@baraodemaua.br (K.F.Z.); 2Departamento de Ciências Farmacêuticas. Faculdade de Ciências Farmacêuticas de Ribeirão Preto, Universidade de São Paulo, Avenida do Café, s/n, Ribeirão Preto, SP 14040-903, Brazil; debora_munhoz@yahoo.com.br (D.M.R.); jkbastos@fcfrp.usp.br (J.K.B.)

**Keywords:** melanoma, skin cancer, baccharin, coumaric acid, cancer therapy

## Abstract

**Background:** Cutaneous melanoma is the most aggressive form of skin cancer, with the worst prognosis, and it affects a younger population than most cancers. The high metastatic index, in more advanced stages, and the high aggressiveness decrease the effectiveness of currently used therapies, such as surgical removal, radiotherapy, cryotherapy, and chemotherapy, used alone or in combination. Based on these disadvantages, research focused on alternative medicine offers great potential for therapeutic innovation. Medicinal plants represent a remarkable source of compounds for the treatment of various diseases. **Methods:** In this study, we investigated the tumoral behavior of melanoma under treatment with the compounds baccharin and p-coumaric acid, extracted from green propolis, in mice inoculated with B16F10 cells for 26 days. **Results:** A significant modulation in the number of inflammatory cells recruited to the tumor region and blood in the groups treated with the compounds was observed. In addition, a significant reduction in the amount of blood vessels and mitosis in the neoplastic area was noticed. **Conclusions:** Through our research, we confirmed that baccharin and coumaric acid, isolated substances from Brazilian green propolis, have a promising anticarcinogenic potential to be explored for the development of new antitumor agents, adhering to the trend of drugs with greater tolerance and biological effectiveness.

## 1. Introduction

Worldwide, more than 200,000 new cases of primary malignant melanoma cancers are diagnosed every year (excluding non-melanoma skin cancers). In addition, about 55,500 deaths are reported annually, corresponding to >80% of deaths from skin cancer and 7% of all cancer deaths [1,2]. Characteristically, cutaneous melanoma is the most aggressive form of skin cancer, with the worst prognosis, and it affects a younger population than most cancers [3,4]. First described by René Laënnac in 1806, cutaneous malignant melanoma is based on the atypical transformation of melanocytes, melanin-producing cells present in the basal layer of the epidermis, that expand into the deep and superficial layers [5]. The risk factors involved in the pathogenesis of this disease are numerous, among those already identified are white skin, hair, and light eyes; sensitivity to the sun; family history of melanoma; and dysplastic nevi. Although sun exposure is considered a great risk factor, areas not exposed to the sun may also develop neoplasias [6]. The melanomas that infiltrate the dermis are capable of producing metastases, and are responsible for the high mortality rate of the disease [4,5,7].

In view of these aspects, in addition to the high metastatic index, in more advanced stages, melanoma has high aggressiveness, which decreases the effectiveness of currently used therapies, such as surgical removal, radiotherapy, cryotherapy, and chemotherapy, used alone or in combination [3,8,9]. In addition, it is worth highlighting the negative aspects related to chemotherapeutic treatment, among them, the serious adverse effects and the induction of resistance to multiple drugs [10]. Based on these disadvantages, research focused on alternative medicine offers great potential for therapeutic innovation. In short, medicinal plants represent a remarkable source of compounds for the treatment of various diseases [11]. Numerous phytochemicals from plants and natural extracts have shown relevant antitumor properties, generating a favorable scenario for the development of new therapies [10].

Popularly known as “Alecrim do campo”, the *Baccharis dracunculifolia* plant, native to the Brazilian south, southeast, and midwest regions, serves as a substrate for the preparation of green propolis. Popularly used since antiquity, propolis is known for its numerous biological and pharmacological properties. In Brazil, its use is widespread for the treatment of tuberculosis, gastric disorders such as duodenal ulcers and fever, and as an anesthetic, antimicrobial, and anticancer agent [12,13,14].

Propolis has remarkable cytotoxicity against several tumor cells, such as human fibrosarcoma cells and human lung adenocarcinoma cells [15]. In the experimental analysis by Akao et al. (2003), baccharin and drupanin extracts, derived from the cinnamic acid present in green propolis and *B. dracunculifolia*, showed antitumor activity on human myelocytic leukemia cells. p-Coumaric acid, a phenolic compound obtained from green propolis, has multiple described biological functions, including antioxidant, antiplatelet, and anticancer activity [16]. Kianmehr et al. (2020) observed that low frequency laser irradiation followed by the use of p-coumaric acid on human melanoma cancer cells reduced the growth and survival of melanoma cells [17]. These findings reinforce the need for research on the subject and better comprehension about these compounds, which have broad therapeutic potential to be explored.

In this study, we investigated the tumoral behavior of melanoma under treatment with the compounds baccharin and p-coumaric acid, extracted from green propolis, in mice inoculated with B16F10 cells. With that, it was possible to observe a significant modulation in the number of inflammatory cells recruited to the tumor region in groups treated with baccharin and p-coumaric acid. There was also a significant reduction in the amount of blood vessels and mitosis in the neoplastic area. These results indicate a promising anticarcinogenic potential for these propolis-derived compounds, which can be explored for the development of new antitumor agents with greater tolerance and biological effectiveness.

## 2. Material and Methods

### 2.1. Baccharin and p-Coumaric Acid Isolation from Green Propolis

The compounds baccharin and p-coumaric acid were obtained from 300 mg of green propolis (supplied by Apis Flora Ltd. Ribeirão Preto, SP, Brazil), which were frozen, pulverized, and extracted with hydroalcoholic solution (9:1). After this process, the extracts were filtered, concentrated under reduced pressure, and lyophilized to obtain the crude hydroalcoholic extract.

Subsequently, the crude extract was dissolved in MeOH/H_2_O and partitioned with hexane (hex), dichloromethane (DCM), and ethyl acetate (EtOAc). The DCM extracts were further fractioned in a classic chromatographic column with gradient elution (silica gel, hex/EtOAc 95:05 to 85:15 *v*/*v*). Each subfraction was analyzed by HPLC-DAD and the rich subfraction in baccharin and p-coumaric acid were injected into preparative HPLC-UV, with a reverse phase column (4 μm, 250 × 10 mm) and detection at 275 nm, to isolate about 50 mg of these two compounds. Baccharin and p-coumaric acid were identified by spectroscopic and spectrometric analyses. After that, the compounds were suspended in 997 μL of 1x PBS (saline phosphate buffer) and 3 μL of DMSO (dimethylsulfoxide) with a final concentration of 1 mg/mL [18].

### 2.2. Animal Care

Male and female Balb/C mice (6–8 weeks of age) of approximately 20 g were obtained from the Centro Universitário Barão de Mauá (São Paulo, Brazil). The animals used in the experiment were kept at 25 °C, with a 12 h/12 h light/dark cycle, with free access to water and food. All experiments were approved in advance and conducted according to the Animal Research and Experimentation Ethics Committee of the Centro Universitário Barão de Mauá (process number: 346/19, accepted 13 March 2019).

### 2.3. Cell Culture

B16F10 melanoma cells were cultured in an appropriate culture medium and, once in the confluent phase and exhibiting exponential growth, they were adjusted to the concentration of 10^6^ cells/mL in 0.1% saline of incomplete DMEN medium (Dulbecco modification of Minimum Essential Media).

### 2.4. Experimental Design

The animals (*n* = 36) used in the experiment were divided into eight groups, four groups per experiment.
Group 1—Control: received Phosphate buffer saline (PBS) v.o; Group 2—PBS + baccharin: PBS injection + treatment with baccharin v.o.;Group 3—PBS + p-coumaric: PBS injection + treatment with p-coumaric acid v.o.;Group 4—Melanoma: Injection of B16F10 cells + PBS v.o.;Group 5—Melanoma + baccharin: Injection of B16F10 cells + treatment with baccharin v.o. Group 6–Melanoma + p-coumaric: Injection of B16F10 cells + treatment with p-coumaric acid v.o. 

Then, the mice had their dorsal region tricomized with the aid of a Panasonic shaver (approximately 1.5 cm in diameter). On the first day of the experiment, the mice belonging to the Melanoma groups (*n* = 7), Melanoma + baccharin (*n* = 7), and Melanoma + p-coumaric (*n* = 7) were inoculated with B16F10 cells in 10^6^ cells/0.1 mL concentration in 0.9% PBS in the dorsal subcutaneous region. The Control (*n* = 5), Baccharin (*n* = 5), and p-coumaric (*n* = 5) groups received 0.9% PBS in the dorsal subcutaneous region. After inoculation of the tumor, the treated groups, Melanoma + baccharin and baccharin, ingested via gavage baccharin compound (500 μg/kg v.o) for 26 days. In the same way, the Melanoma + p-coumaric acid and p-coumaric acid groups, also ingested p-coumaric compound (500 μg/kg v.o) via gavage for 26 days. The Control and Melanoma groups received PBS (1 mL v.o.) via gavage for 26 days (Figure 1a). During the experiment, survival was monitored. At the end of the treatment, the animals were sacrificed in a CO_2_ chamber 30 days after the tumor was inoculated.

### 2.5. Inflammatory Cells Counting in Peripheral Blood 

For the collection of blood samples, the mice were anesthetized with Ketamine/xylazine through the retro-orbital plexus with the help of a Pasteur pipette containing heparin. Blood cell counting was performed in a Turk solution using a Neubauer chamber. Differential leukocyte counting was done on spotted cytosine preparations using a commercial Romanowsky procedure kit. One hundred cells were counted in random fields in a 100x magnification under the light microscope.

### 2.6. Analysis of Tumor Growth 

A pachymeter was used to measure the dimensions of the tumor (in mm³) every 5 days, throughout the experiment, being recorded in such a way that tumor volume = (length × height × width)/2.

### 2.7. Histological and Morphometric Evaluation

For histological evaluation, tumor samples were collected with an adjacent tissue margin to preserve the microenvironment and fixed in a 10% formaldehyde solution. They were dehydrated with ethyl alcohol in an increasing series (70%, 80%, 90%, and 100%). The diaphanization process used xylol as a reagent. Paraffin blocks after preparation were sectioned (4 μm) and stained with hematoxylin and eosin (HE). For the morphometric evaluation, a Nikon eclipse E200 microscope coupled to a video camera (Tucsen USB 2.0 H Series) was used to analyze the morphometry. The primary tumor vessels were quantified at 40× magnification in 15 random non-coincident microscopic fields. Inflammatory cells (macrophages and neutrophils) were quantified at 400× magnification in 15 random non-coincident microscopy fields. For this, the ISC program, Tucsen Photonics Co., Ltd. (Fuzhou, China), was used.

### 2.8. Statistical Analysis 

Using the GraphPad v 8.0 software (GraphPad, San Diego, CA, USA), statistical analysis of the obtained data was performed. Data were expressed as mean +/− standard error of the mean. The Student’s *t*-test was used to assess the differences between any two groups, and for the comparison of multiple groups, the ANOVA test (unidirectional analysis of variance) was performed followed by the Tukey’s test. Values of *p* < 0.05 were considered significant.

## 3. Results

### 3.1. Tumor Growth and Monitoring of Mice 

Two independent experiments were performed with two different compounds: baccharin and p-coumaric acid. In both trials, Balb/C mice had their tumor growth monitored for 26 days after the injection of B16F10 melanoma cells. During this period, 6 measurements of the volume (mm³) were obtained using a pachymeter. At the end of the experiment, the regions that showed tumor growth and melanin accumulation were extracted and analyzed. The survival rate of the animals showed no significant difference during both experiments between the groups, considering that there were no deaths in any group in the period. In the baccharin test, the values showed no difference between the means of the ANOVA with Tukey’s post-test. In addition, the group treated with p-coumaric acid showed a significant drop in tumor volume (236.4 ± 89.64), suggesting that the compound is capable of controlling tumor growth (Figure 1).

### 3.2. Effects of Baccharin and p-Coumaric Acid on Mitotic Cells

The number of mitosis events is a determining factor for understanding the evolution and progression of a tumor, since high cell growth rates give to melanoma the ability to develop and invade new adjuvant tissues. Based on this concept, the number of mitotic cells was quantified in histological fragments obtained from animals with B16F10 melanoma cells implanted subcutaneously, and that were treated or not for 26 days. Non-coincident microscopy fields were analyzed. At first, inoculation of B16F10 cells induced a significant increase in mitosis. As a result, the group that received treatment with baccharin (500 µg/kg p.o.) during the study period had a significant reduction in the number of cells in mitotic activity observed in the tumor region. We observed that the Melanoma + baccharin group had an average of mitoses in the lower tumor region (1.567 ± 0.2554) compared to that of the Melanoma group (3.486 ± 0.2554) (*p* < 0.0001). Similarly to baccharin, the group treated with p-coumaric also showed a significant difference in relation to the rate of mitosis. A reduction in the number of mitoses per field was observed (1.356 ± 0.2475) when compared to that of the Melanoma group (3.486 ± 0.2475) *(p* < 0.0001), suggesting that the presence of the compound influences this point (Figure 2). These differences were considered significant with *p* < 0.05 according to the Student’s *t*-test and ANOVA for both experiments. 

### 3.3. Effects of Baccharin and p-Coumaric Acid in Tumor Angiogenesis

Angiogenesis is the formation of new blood vessels. This process, when targeted at tumor development, allows cancer cells to establish themselves at the destination site and to maintain continuous growth, ensuring nutrition and a gateway to mitoses. In our study, we quantified the number of vessels per optical microscopy in non-coincident fields. From the data obtained and submitted to analysis, we observed that in the animals belonging to the Melanoma group, the average number of vessels per field (4.8 ± 0.347) under optical microscopy was higher than that of the animals of the Melanoma + baccharin group (3.1 ± 0.347) (*p* < 0.0001). There was a significant difference between the formation of vessels in the groups treated with baccharin. Similarly, in the Coumaric test, a significant reduction in the number of blood vessels (2.367 ± 0.3243) (*p* < 0.0001) was reported when compared to the untreated group (Melanoma). Baccharin and p-coumaric acid showed a positive performance by reducing the formation of new vessels in the experiment (Figure 3). These differences were considered significant with *p* < 0.05 according to the Student’s *t*-test and ANOVA test.

### 3.4. Effects of Baccharin and p-Coumaric Acid in Inflammatory Tissue Cells 

Inflammation is a process involved in the pathogenesis and progression of several diseases, including cancer. Therefore, we sought to determine the number of inflammatory cells present in the skin fragments of melanoma extracted from mice by observing and quantifying neutrophils and macrophages. The tumor cells induced a significant increase in the neutrophils recruited to the region observed in the optical microscopy non-coincident fields of the histological fragments (Figure 4). Taking into account these parameters, there was no statistical difference between the Melanoma group and the Melanoma + baccharin group. However, the group with Melanoma + p-coumaric compound showed a significant reduction (17.09 ± 2.235) *(p* < 0.0001) in the recruitment of these cells to the injured area, when compared to that of the Melanoma Group (26.30 ± 2.235). This suggests that the p-coumaric compound acts better in the acute inflammation phase of the disease. These differences were considered significant with *p* < 0.05 according to the Student’s *t*-test and ANOVA test.

When activated by a combination of factors, macrophages can kill specific tumor cells. In certain circumstances, macrophages may present tumor-associated antigens on T cells and stimulate the tumor-specific immune response [19]. Based on this, we observed that the average number of macrophages by non-coincident field in the Melanoma group (2.42 ± 0.39) was considerably lower if compared to that of the Melanoma + baccharin group (3.59 ± 0.39), resulting in a difference of statistically relevant means. These differences were considered significant with *p* < 0.05. Based on the same analysis criteria, it was observed that the macrophages count between the groups Melanoma and Melanoma + p-coumaric had no significant difference (Figure 5). 

### 3.5. Effects of Baccharin and p-Coumaric Acid on Inflammatory Cells in Blood

As expected, analysis of peripheral blood collected after the animals’ sacrifice showed a significant increase in total leukocytes in animals that were inoculated with B16F10 melanoma cells. However, there was no statistically significant difference between the Melanoma group and either the Melanoma + baccharin group or Melanoma + p-coumaric group. When analyzed, the number of mononuclear cells also did not show any statistical difference between all groups under tumor conditions.

It was observed that the animals inoculated with B16F10 cells that received the baccharin treatment during the period of experiment, Melanoma + baccharin group (919.2 ± 0.002), had a significant reduction in relation to the number of neutrophils when compared to that of the Melanoma group (1851 ± 0.002) (*p* = 0.0372). Similarly, the neutrophils in the group that was inoculated with the B16F10 melanoma cells and treated with p-coumaric showed a significant drop in the counting of these cells (17.09 ± 2.376) (*p* = 0.0001) when compared to that of the other groups (Figure 6). These differences were considered significant with *p* < 0.05.

Under physiological conditions, both baccharin and p-coumaric acid did not change the number of total leukocytes, mononuclear cells, or neutrophils quantified in peripheral blood.

## 4. Discussion

Despite constant advances and research in the field of oncology, treatment for melanoma remains challenging. In addition to the low effectiveness of chemotherapy drugs available at advanced levels of metastasis, one of the main negative aspects of chemotherapy is its serious adverse effects and the induction of resistance to multiple drugs [10]. In fact, drugs are differently processed and metabolized in patients, possibly modifying both effectiveness and toxicity of treatments [20]. Due to the disadvantages related to conventional chemotherapy, research with natural compounds represents a great potential for innovation [10,21,22]. Thus, the objectives are focused on identifying drugs that act in specific target sites while also being bioavailable and non-toxic [7]. Through this study, we investigated the tumor development induced by melanoma cells B16F10 and the anticarcinogenic properties of baccharin and p-coumaric acid, isolated compounds from green propolis, in mice treated orally for 26 days. Previous in vitro studies demonstrated the cytotoxic effects of propolis extracts as well as of isolated specific compounds in cell lines derived from different cancer types, such as breast [23,24], colon [25], uterine, cervix, and lung [26]. Accordingly, we aimed to obtain an in vivo proof of concept about the action of baccharin and p-coumaric acid on certain fundamental factors related to the development and progression of tumors. Our research group has shown promising results regarding the therapeutic properties of natural extracts. Nascimento et al. (2019) observed favorable results in an experimental model of acute inflammation and tumor progression with the extract of the *Arctium lappa* plant, demonstrating that in addition to its direct effect on tumor cells, *A. lappa* extract suppresses the migration of neutrophils, which is correlated with lower levels of cytokines such as IL-6, IL-1 and TNF-α [27]. According to Garcia et al. (2020), the hydroalcoholic extract of *Lobelia inflata* contains compounds with significant antitumor activity against melanoma, with reduced recruitment of inflammatory cells, edema, and cytokines in the peritoneal cavity [28]. From the perspective of our research, the study described by Ireno et al. (2020), which also focused on the *Lobelia inflata* extract on melanoma, showed a reduction in the angiogenesis process due to the decrease in blood vessels after interaction with the extract [29]. Thus, these findings obtained from different natural extracts encourage the development of new studies for the treatment of melanoma.

Propolis has activity against several tumor cells in vitro and in vivo. In short, due to its immunomodulatory action based on the nonspecific antitumor immunity response via macrophage activation, it produces soluble factors and acts directly on tumor cells or in the function of other cells [30]. It has been demonstrated that propolis can disrupt oncogenic signaling pathways, inhibit cell growth and proliferation, induce apoptosis, and reduce angiogenesis [31,32,33], among other effects [34,35]. In our study, the antiproliferative activity of constituents isolated from propolis on tumor cells was evaluated. Derivatives of cinnamic acid (baccharin and drupanin), flavonoids, and other phenolic compounds were evaluated on leukemia cells (HL-60), which demonstrated its in vitro cytotoxic effect on this tumor type [36]. Assumpção et al. (2020) selected three phenolic acids (caffeic, dihydrocinnamic, and p-coumaric) commonly detected in propolis to evaluate if they could act as epigenetic drugs, reverting the acquired epigenetic changes associated with tumor resistance to therapy on four triple-negative breast cancer cells lines. Among the tested phenolic acids, only p-coumaric showed cytotoxic effects [37]. In summary, the main mechanisms involved in the action of green propolis and its compounds are related to the inhibition of cell growth during mitosis, apoptosis, and interference in the metabolic pathways [30,38,39]. The apoptosis mechanism induced by propolis does not depend on the type of cancer cell, but on the concentration of propolis. Studies available in the literature indicate that propolis induces apoptosis by releasing cytochrome c from the mitochondria to cytosol, through the cascade of caspases and TRIAL (TNF-related apoptosis-inducing ligand) signal. Concerning tumor proliferation, it is also known that propolis can inhibit the cell growth mechanism in two ways: (1) stopping proliferation in the G2 phase of mitosis, thereby preventing the cell from advancing to the next phase if division occurs and (2) decreasing the activity of telomerase, an enzyme that acts to protect the telomeres, preventing the death of these defective cells. Based on this information, the following paragraphs are dedicated to comparing data available in the literature with our findings about baccharin and p-coumaric acid’s action on the development of melanoma in mice.

Cell proliferation in the primary tumor, as reflected by its mitotic rate, is an important predictor of survival [40,41]. Many mitoses in any primary tumor indicate that cells are actively dividing and these tumors tend to grow more quickly and can metastasize earlier [40,42]. Propolis and its compounds induce the inhibition of cell proliferation by suppressing complexes of cyclins and cyclin-dependent protein kinases, in addition to raising the levels of protein inhibitors, such as p21, p16, and p27 in tumor cells, and inducing the cell cycle arrest [39]. The histological analysis of a study carried out by Kuo et al. (2006) demonstrated that caffeic acid phenethyl ester (CAPE), a component of propolis, when used on glioma cells, decreased the number of mitoses [43]. In this regard, our results showed an important decrease in the number of tumor mitoses observed by microscopic field in animals treated with baccharin and p-coumaric acid compounds. When evaluated, mice that received the compounds for 26 days had a reduction of more than 50% in the number of mitoses compared to that of untreated animals. These data indicate that baccharin and p-coumaric acid have anti-mitotic capacity on B16F10 melanoma cells and consequently influence the tumor progression in mice. Reinforcing this analysis, Boo et al. (2019) mentions that p-coumaric acid has been shown to inhibit proliferation and migration of cancer cells and promote apoptotic cancer cell death, supporting its potential anticancer effects [44]. 

Angiogenesis, the process of forming new vessels, is an important step towards tumor development [45,46]. This mechanism is promoted by the expression of endogenous angiogenic factors, such as interleukin-8 and VEGF (Vascular Endothelial Growth Factor), which lead to an increase in the local formation of new vessels [47]. Melanoma cells produce numerous cytokines and growth factors. Many of these growth factors act in synergy with extracellular matrix receptors, integrins, and metalloproteinases. All of these molecular systems are activated during the stages of angiogenesis: endothelial migration, proliferation, and reorganization of the surrounding extracellular matrix. Through this process, melanoma ensures its growth and metastasis [47,48,49]. In view of these aspects, inhibiting angiogenesis can be a mechanism of choice for treating tumors, acting on microenvironmental factors and endothelial containment [46]. In our results we observed that the animals treated with baccharin and p-coumaric acid had a reduction in tumor angiogenesis greater than 30% and 50%, respectively. These data represent an important aspect about tumor inhibition, considering that tumor nutrition and metastasis are associated with blood vessels. In the study described by Watanabe et al. (2011), the production of antiangiogenic factors was observed in animals inoculated with B16F10 melanoma cells when treated with Brazilian propolis extract. This result was attributed to a synergistic effect between pro-inflammatory cytokines and interferon-γ, whose increased levels were observed in animals that received propolis [30]. Lima et al. (2014) had similar results showing a decrease in an experimentally implanted neovascularization after treatment with Brazilian green propolis [50]. 

Inflammation is a process involved in the pathogenesis and progression of several diseases. Usually, the inflammatory reaction seeks to restore homeostasis affected by an injury or infection. In tumor development, the immune process is a fundamental component that can follow two routes: tumor protection and inhibition or facilitation of neoplastic development [51]. At the beginning of the neoplastic process, inflammatory cells can be tumor facilitators, producing an attractive environment for growth, hastening genomic instability, and promoting angiogenesis. The tumor process diverts inflammatory mechanisms, such as selectin–ligand interactions, MMP (metalloproteinases) production, and chemokine functions. However, the recruitment of inflammatory cells also represents a counter-tumor mechanism, being an attempt by the host to suppress tumor growth [9]. Genetic changes associated with cancer result in the expression of tumor-associated antigens (TAAs) and tumor-specific antigens (neoantigens). These antigens can activate anti-tumor immunity and also induce rejection of early neoplasms, acting as a tumor signal for the immune system to attack [52].

The ability of active macrophages to destroy tumor cells in vitro, such as gliomas, has long been known [53,54]. The mechanisms involved in the action of macrophages against tumor cells are the production of NO (nitric oxide) and TNF-α secretion, acting directly on tumor cells. Besides that, tumor-associated macrophages (TAM) can be polarized upon IFN-γ stimulation into a M1 phenotype, which exhibits enhanced anti-tumorigenic properties, or into a M2 phenotype upon IL-4 stimulation, which exhibits pro-tumorigenic activities [55]. Based on the results observed in our study, baccharin induces an activating response on macrophages, considering that the treated animals had an increase of macrophages on the tumor site when compared to that of animals that did not receive the compound during the experiment. Previous findings suggest the activating action of macrophages in the extract and compounds isolated from *B. dracunculifolia* and green propolis [56]. Nonetheless, the experiment with p-coumaric acid did not show the same results about macrophage activation, considering that there was no difference in the recruitment of macrophages to the injured area in animals that were treated or not with p-coumaric acid. In fact, the mechanisms involved in these different performances are not known, since both compounds were derived from the same plant and were submitted to the same experimental condition. Thus, more studies need to be performed. According to the study described by Orsolic et al. (2004), the antitumor property of propolis and some of its constituents can be associated with in vivo immunomodulatory action, mainly by the activation of macrophages that can produce soluble factors and interfere directly with tumor cells or immune cell functions, including the ability to produce apoptosis and/or tumor necrosis [57]. In addition, factors related to the tumor microenvironment and secretion of chemokines and MIF (macrophage migration inhibitory factor), a pro-inflammatory cytokine that inhibits the random movement of macrophages, could also be associated with this difference [58]. 

Recent evidence shows that activation of neutrophils can lead to tumor progression and metastasis through stromal remodeling, stimulate angiogenesis, or compromise T-cell-dependent antitumor immunity [59,60,61]. Fridlender et al. (2009) have shown that neutrophils can also be polarized into an anti-tumorigenic N1 phenotype following IFN-β stimulation or into a pro-tumorigenic N2 phenotype following transforming growth factor beta stimulation [62]. There is also increasing evidence in both human and murine studies that neutrophils inside the tumor are predominantly pro-tumorigenic, especially in advanced and established tumors [61,63]. However, in early tumors, tumor-associated neutrophils (TAN) are more cytotoxic and produce higher levels of TNF-α, NO, and H_2_O_2_. In our study we observed a reduction of neutrophils at the damaged tissue site of animals treated with p-coumaric acid. In this sense, the animals that received the compound had a decrease in recruiting neutrophils compared to that of untreated animals, which is a good find, since Jensen et al. (2012) revealed that the presence of intratumoral neutrophils significantly correlated with poor survival in primary melanomas from patients [64]. Supporting this find, the literature shows us that phenolic compounds, such as p-coumaric acid, have reducing power on TNF-alfa levels and free radical scavenger activity and may be helpful in the prevention or alleviation of many chronic diseases caused by oxidative stress [65,66,67]. In the same way, our study also showed a reduction of neutrophils in the blood of animals treated with baccharin and p-coumaric acid. In this sense, the animals that received these compounds had a reduction of more than 50% in the number of neutrophils circulating in the bloodstream compared to that of untreated animals, in both experiments. Tan et al. (2017) showed an increase of neutrophils in peripheral blood and also correlated this with poor survival and non-responsiveness to ipilimumab, a human monoclonal antibody used to treat melanoma [68]. Although peripheral blood levels of immune cells can be useful biomarkers of disease progression, the immune cells need to be physically in the tumor to exert their functions [68].

In the present study we analyzed the effects of two compounds from green propolis in melanomas developed in mice. Research on mice has contributed vastly to our understanding of human biology; however, mice can respond to experimental interventions in ways that can differ from humans. The anti-melanoma therapy based on green propolis (baccharin and p-coumaric) has not yet been validated in humans.

## 5. Conclusions

The present study concluded that baccharin and p-coumaric acid, isolated substances extracted from Brazilian green propolis, act on certain factors that contribute to tumor growth and progression, such as angiogenesis, number of mitoses, and modulatory activity on certain inflammatory cells, including macrophages and neutrophils. It is worth mentioning that we provide evidence about their activity in mice. However, future studies are necessary to delimit their therapeutic effects based on the understanding of their mechanisms of action.

## Figures and Tables

**Figure 1 medicines-08-00020-f001:**
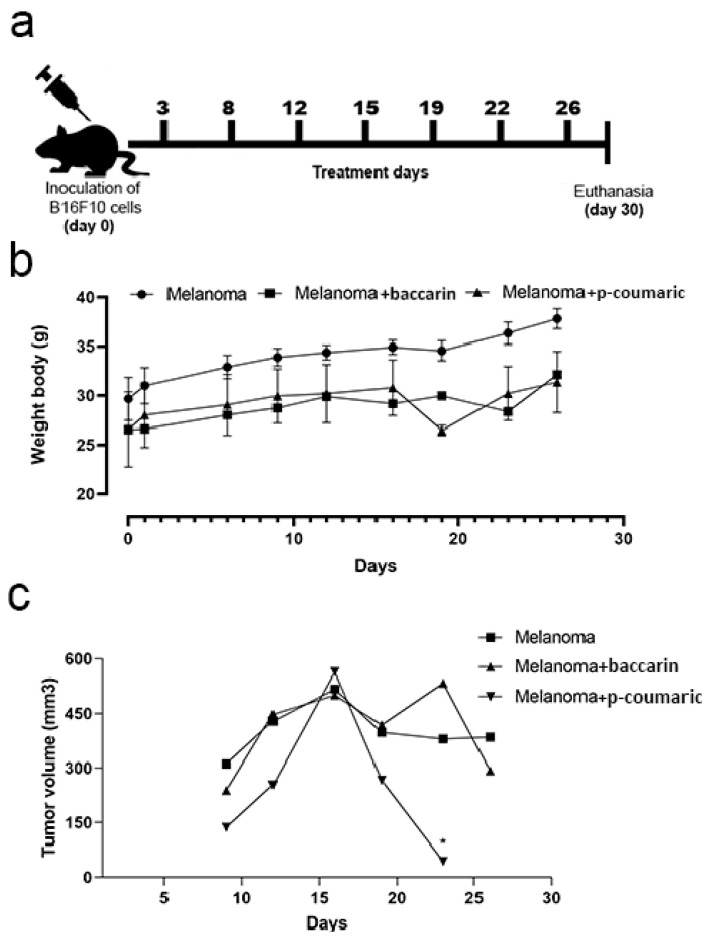
(**a**) Illustration of animal treatment days. (**b**) Representative graphic of body weight (g) during the experiment. (**c**) Representative graphic of the tumor volume (mm³) analysis. Comparison between the experiments with baccharin and p-coumaric acid: Melanoma group (B16F10 concentration of 10^6^ cell/mL and ingested PBS) and Melanoma + baccharin group that received treatment with baccharin compound (500 µg/kg). Melanoma group (B16F10 concentration of 10^6^ cell/mL and ingested PBS) and Melanoma + p-coumaric acid group that received treatment with p-coumaric compound, both for 26 days. ***** Data represents mean ± SEM.

**Figure 2 medicines-08-00020-f002:**
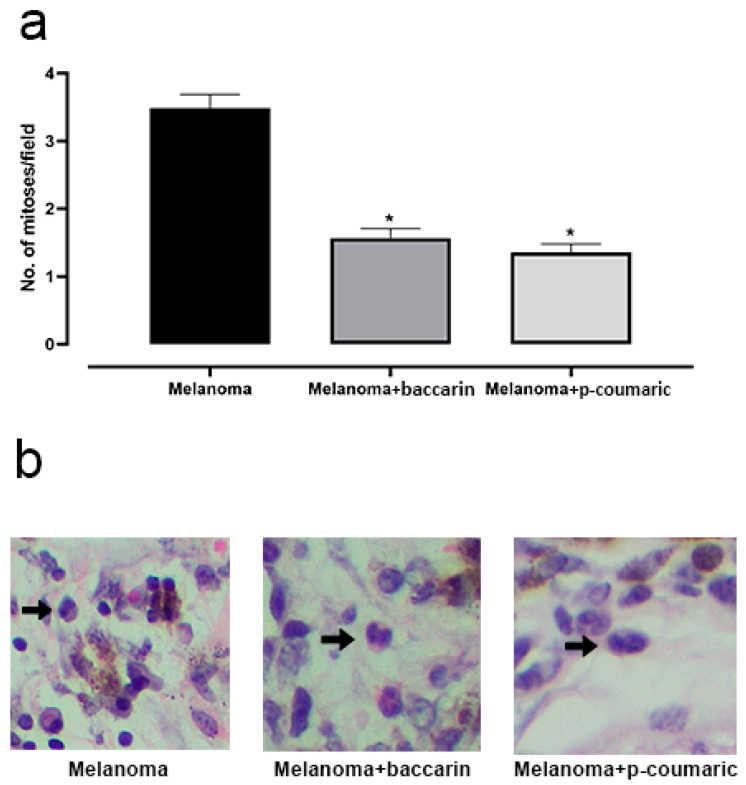
(**a**) Representative graphic of the mitosis analysis. Baccharin and p-coumaric acid reduced tumor mitotic activity. Comparison between the Control and Melanoma groups (B16F10 concentration of 10^6^ cell/mL) that ingested only PBS and the baccharin and Melanoma + baccharin groups that received treatment with the baccharin compound (500 µg/kg) for 26 days. Comparison with the Control and Melanoma groups (B16F10 concentration of 10^6^ cell/mL) that ingested only PBS and the p-coumaric acid and Melanoma + p-coumaric acid groups that received the treatment with p-coumaric compound (500 µg/kg) for 26 days. (**b**) Representative images of the mitosis in 200× magnification. Data represents mean ± SEM. * These differences were considered significant with *p* < 0.05.

**Figure 3 medicines-08-00020-f003:**
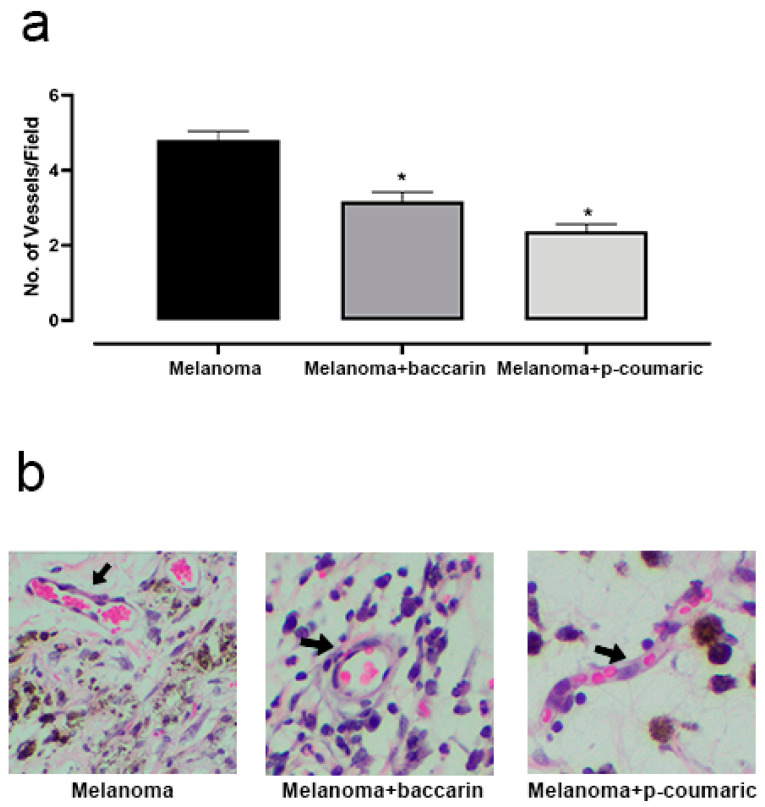
(**a**) Representative graphic of the vessel’s analysis. Baccharin and p-coumaric acid reduces the process of angiogenesis. Comparison between the Melanoma groups (B16F10 concentration of 10^6^ cell/mL) that ingested only PBS and the Melanoma + baccharin group that received treatment with baccharin compound (500 µg/kg) for 26 days. Comparison with the Melanoma group (B16F10 concentration of 10^6^ cell/mL) that ingested only PBS and Melanoma + p-coumaric acid group that received treatment with p-coumaric compound (500 µg/kg), for 26 days. (**b**) Representative images of the histological section of blood vessels in 200× magnification. Data represents mean ± SEM. * These differences were considered significant with *p* < 0.05.

**Figure 4 medicines-08-00020-f004:**
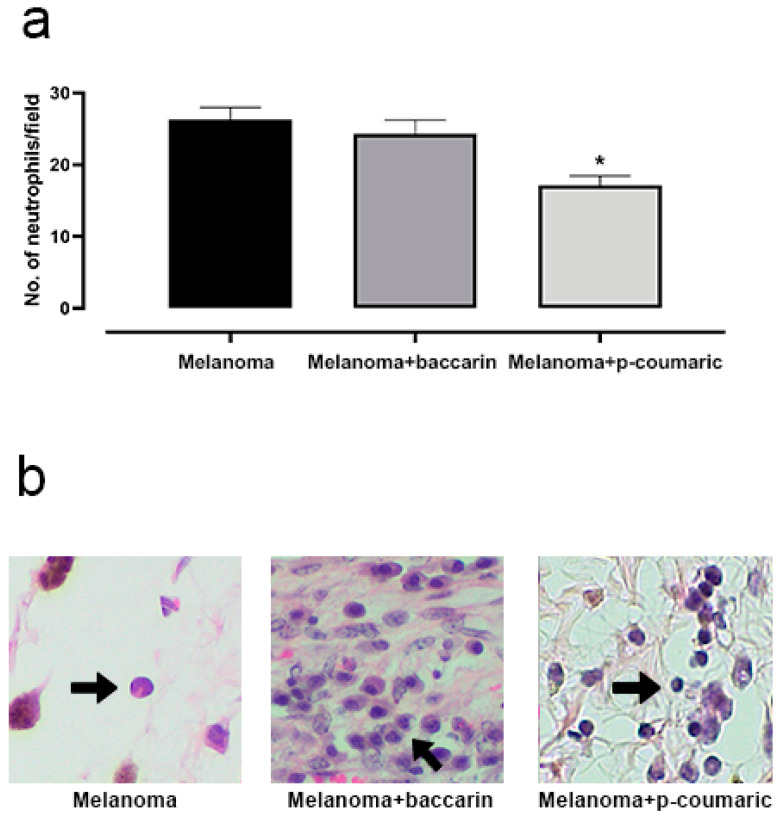
(**a**) Representative graphic of the tumor neutrophils analysis. The tumor induces an increase in neutrophils. Comparison between the Control and Melanoma groups (B16F10 concentration of 10^6^ cell/mL) that ingested only PBS and the baccharin and Melanoma + baccharin, who received treatment with the baccharin compound (500 µg/kg) for 26 days. However, in the comparison between the Control and Melanoma groups (B16F10 concentration of 10^6^ cell/mL) that ingested only PBS and the p-coumaric and Melanoma + p-coumaric acid groups that received treatment with the compound for 26 days, there was a significant reduction (*p* < 0.05) in the recruitment of these cells. (**b**) Representative images of the histological section in 200x magnification. Data represents mean ± SEM. * These differences were considered significant with *p* < 0.05.

**Figure 5 medicines-08-00020-f005:**
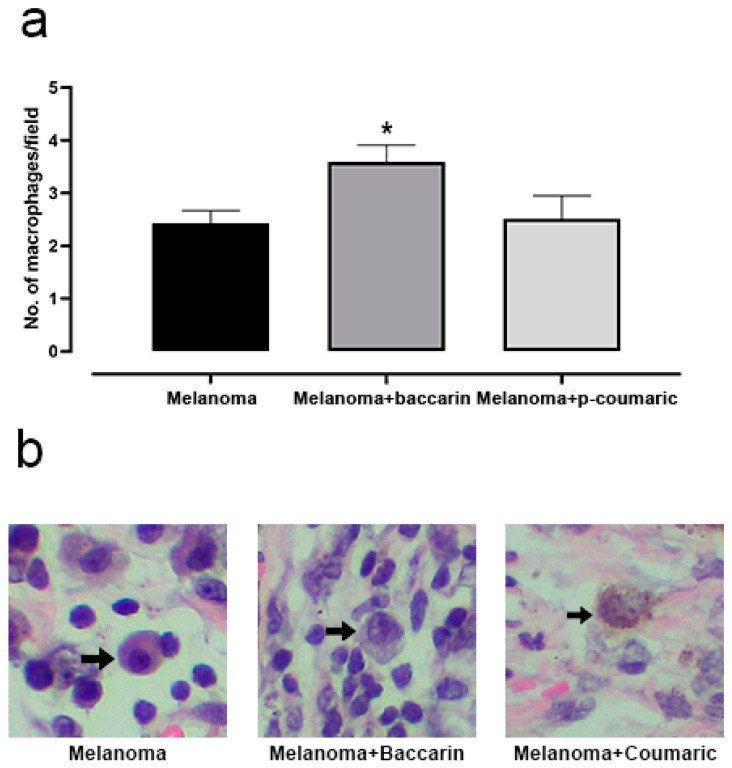
(**a**) Representative graphic of the tumor macrophages analysis. Baccharin increases the number of macrophages recruited to the tumor region. Comparison between the Melanoma group (B16F10 concentration of 10^6^ cell/mL) that ingested only PBS and the Melanoma + baccharin group that received treatment with baccharin compound (500 µg/kg) for 26 days. However, the p-coumaric compound did not show an effect on the recruitment of macrophages cells to the injured area. Comparison between the Melanoma group (B16F10 concentration of 10^6^ cell/mL) that ingested only PBS and Melanoma + p-coumaric acid group that received the treatment with p-coumaric compound (500 µg/kg), for 26 days. (**b**) Representative images of the macrophages’ histological section in 200× magnification. Data represents mean ± SEM. * These differences were considered significant with *p* < 0.05.

**Figure 6 medicines-08-00020-f006:**
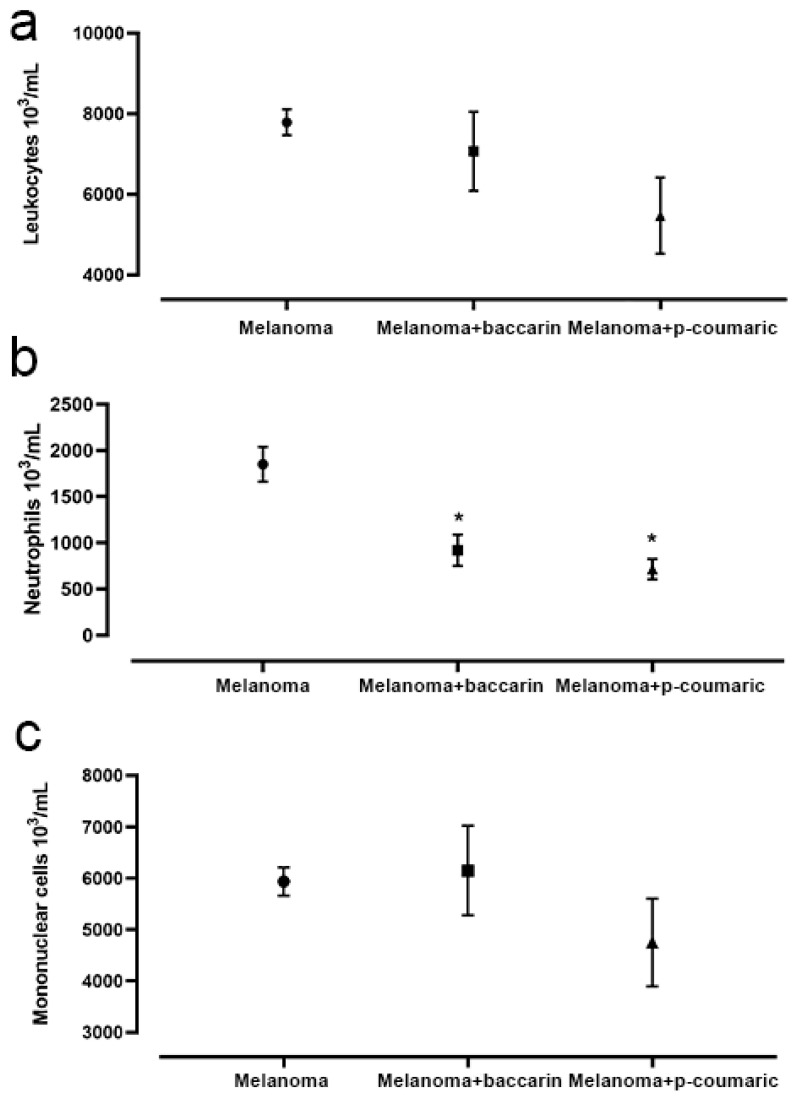
Representative graphic of the analysis of total leukocytes (**a**), neutrophils (**b**), and mononuclear cells (**c**) in peripheral blood. There were no significant differences in the analysis of total leukocytes and mononuclear cells between the experiments under tumor conditions. However, there was a reduction in the number of neutrophils in the Melanoma + baccharin and Melanoma + p-coumaric acid groups compared to that of the Melanoma group. Comparison between the Control and Melanoma groups (B16F10 concentration of 10^6^ cell/mL) that ingested only PBS and the baccharin or p-coumaric and Melanoma + Baccharin groups that received treatment with the baccharin compound (500 µg/kg) for 26 days and the Melanoma + p-coumaric acid group that received treatment with p-coumaric compound (500 µg/kg) for 26 days. Data represents mean ± SEM. * These differences were considered significant with *p <* 0.05.

## Data Availability

Not applicable.

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
