# Peer review of "Green Propolis Compounds (Baccharin and p-Coumaric Acid) Show Beneficial Effects in Mice for Melanoma Induced by B16f10"

_medicines, 2021, doi:10.3390/medicines8050020_

Round 1
Reviewer 1 Report
I was glad I was called to review this manuscript.Overall it is well written and argued with a good introduction. As far as Materials and Methods are concerned, I believe that the images of mitosis should be revised, as it is difficult to visually appreciate the mitotic figures (Figure 2b). For the rest it seems to me a well written paragraph.
In figure 3b I would suggest that the authors implement the images with immunohistochemical staining for CD34 / CD31 to clearly graphically document the reduction of blood vessels described in the article.
In figure 4b I believe it would be more complete and effective to perform immunohistochemical staining for CD55.As for the "Results" secI would suggest to the authors to perform a Ki67 + to more adequately evaluate the impact of the treated compounds on the neoplastic proliferation index, as well as the simple mitotic index. Furthermore, in Figure 5b I suggest to perform an immunohistochemical evaluation for CD68, in order to evaluate macrophages more precisely. As for the "Results" section, I would suggest to the authors to perform a Ki67 + to more adequately evaluate the impact of the treated compounds on the neoplastic proliferation index, as well as the simple mitotic index. Furthermore, in Figure 5b I suggest to perform an immunohistochemical evaluation for CD68, in order to evaluate macrophages more precisely. As far as the discussion is concerned, I believe that it is based on a good basis, but I would emphasize the fact that the results obtained (the authors clarify this only at the end in the "Conclusions") are related to experiment mice, so I would be very cautious in stating that this launches huge promises for new therapies in humans. No human studies have yet been validated, so although well written and with good evidence, it remains an animal study. I would recommend rephrasing some statements in light of these highlighted limitations.
Author Response
Reviewer1
Overall it is well written and argued with a good introduction. As far as Materials and Methods are concerned, I believe that the images of mitosis should be revised, as it is difficult to visually appreciate the mitotic figures (Figure 2b). For the rest it seems to me a well written paragraph.
In figure 3b I would suggest that the authors implement the images with immunohistochemical staining for CD34 / CD31 to clearly graphically document the reduction of blood vessels described in the article.
In figure 4b I believe it would be more complete and effective to perform immunohistochemical staining for CD55.As for the "Results" secI would suggest to the authors to perform a Ki67 + to more adequately evaluate the impact of the treated compounds on the neoplastic proliferation index, as well as the simple mitotic index. Furthermore, in Figure 5b I suggest to perform an immunohistochemical evaluation for CD68, in order to evaluate macrophages more precisely. As for the "Results" section, I would suggest to the authors to perform a Ki67 + to more adequately evaluate the impact of the treated compounds on the neoplastic proliferation index, as well as the simple mitotic index. Furthermore, in Figure 5b I suggest to perform an immunohistochemical evaluation for CD68, in order to evaluate macrophages more precisely.
Reply: Thank you for this suggestion. This study aims to compare two different compounds (baccarin and p-coumaric) showing the morphological analysis. Initially, the propose is to demonstrate morphological changes. In this sense, a more detailed analysis of the aspects raised in this study, including the use of immunohistochemical markers to better elucidate some data, is an objective that we intend to achieve soon. Our research group is enthusiastic about the advances in this line of research and we intend, at the we have resources, make it more comprehensive and diverse. Once again, I thank you for the suggestion, emphasizing that initially, we had as objective a general morphological analysis, and with the data available, we intend to refine these concepts.
As far as the discussion is concerned, I believe that it is based on a good basis, but I would emphasize the fact that the results obtained (the authors clarify this only at the end in the "Conclusions") are related to experiment mice, so I would be very cautious in stating that this launches huge promises for new therapies in humans. No human studies have yet been validated, so although well written and with good evidence, it remains an animal study. I would recommend rephrasing some statements in light of these highlighted limitations.
Reply: As suggested by the reviewer, a paragraph about the limitations of study in mice was addicted.
Reviewer 2 Report
An article by Gastaldello et al. describes green propolis compounds (bacarine and p-coumaric acid) with beneficial effects in mice caused by B16F10 melanoma. In my opinion, the authors proved that the tested propolis components are active in mice. The article can be accepted in its current form.
Author Response
Reply. Thanks for the appreciation of our article.
Round 2
Reviewer 1 Report
The authors responded to all my requests. For me it can be accepted.